# Fatigue Performance of Laser Welds in Heavy-Gage Press Hardening Steels

Diego Tolotti de Almeida [1,2,*], Kleber Eduardo Bianchi [3], João Henrique Corrêa de Souza [3], Milton Sergio Fernandes de Lima [4], Thomas Gabriel Rosauro Clarke [1], Fabio Pinto da Silva [5] and Hardy Mohrbacher [6,7,*]

1   Physical Metallurgy Laboratory/LAMEF–Federal University of Rio Grande do Sul (UFRGS), Porto Alegre 91501-970, Brazil; thomas.grclarke@gmail.com
2   Bruning Tecnometal Ltda, Panambi 98280-000, Brazil
3   School of Engineering, Federal University of Rio Grande, Porto Alegre 96203-900, Brazil; bianchi.kleber@gmail.com (K.E.B.); joaoh_cs@hotmail.com (J.H.C.d.S.)
4   Institute for Advanced Studies/IEAV, Technological Institute of Aeronautics/ITA, São José dos Campos 12228-900, Brazil; msflima@gmail.com
5   Materials and Design Selection Lab, Federal University of Rio Grande do Sul (UFRGS), Porto Alegre 90040-020, Brazil; fabio.silva@ufrgs.br
6   Department of Materials MTM, Leuven University (KU Leuven), 3001 Heverlee, Belgium
7   NiobelCon BV, 2970 Schilde, Belgium
*   Correspondence: diegot@bruning.com.br (D.T.d.A.); hm@niobelcon.net (H.M.); Tel.: +55-9-9722-6045 (D.T.d.A.); Tel.: +32-3-4845260 (H.M.)

**Abstract:** This work investigates and compares the fatigue performance of laser-welded joints of two press hardening steels: a standard 22MnB5 and a variant modified by a combination of niobium and molybdenum (NbMo) alloying. The results indicate that joint geometry aspects, superposed to an intrusion-generated damage mechanism, were prevalent in causing a poor fatigue life in the case of peak stress values greater than 11% of the base metal's ultimate strength being around 1450 MPa. As identical process procedures were employed, the tests allowed investigating the influence of the alloy design on fatigue performance. The results of geometrical analysis and fatigue tests indicated that the NbMo modified alloy performed better than the standard 22MnB5 steel. The results also suggest that, if extremely tight quality limits are used in the manufacturing procedures, laser-welded joints of press hardened steels could offer a very favorable fatigue performance being considerably better than that of conventional and high strength structural steels.

**Keywords:** press hardening steel; laser welding; fatigue behavior; microalloying; heat affected zone microstructure

## 1. Introduction

The transportation industry is continuously searching for better performing materials and design alternatives capable of providing lighter structures, which significantly contribute to reducing the energy consumption during driving operations as well as enabling a higher ratio of payload to vehicle dead weight. Several material options enabling weight reduction including composites and light metals have been evaluated over the last decades. However, up to today steel is still the material of choice in most cases, because of its outstanding mechanical strength and its excellent joinability. Nowadays, specific steel alloys, produced with accurate chemical composition and tight processing control, offer a wide range of possibilities. Some of these alloys, despite presenting a very high strength capacity, still endure plastic deformations before fracture, which is an important characteristic in the case of impact loads and crack-related damage processes [1,2].

Amongst the group of advanced alloys, press hardening steels with a fully martensitic microstructure provide the highest mechanical strength potential. Components made by

the press hardening process and achieving an ultimate tensile strength of 1400–1500 MPa are nowadays widely used in critical body-in-white parts especially when crashworthiness and weight reduction are key design requirements. However, in the case of agricultural and commercial vehicle applications, the involved structures have to endure long and extremely varied dynamic service loads so that fatigue emerges as the prevalent failure mechanism [3,4]. The employed component wall gages are considerably thicker than in the case of light vehicle body-in-white (BIW) components, mainly due to stiffness requirements. The thicker wall gage of these components is also requiring different joining techniques as in BIW applications.

The mechanical performance of a welded structure depends on several factors, especially in the case of a fatigue damage process [5–11]. The fact that welded joints are considered as the weakest link in a structure, especially regarding fatigue performance, is usually addressed by a cautious design approach of positioning these joints in areas of low stress [12]. Typical arc welding processes have been adopted for joining ultra-high-strength steel parts. However, the substantial heat input results in pronounced modification of the microstructure and mechanical properties [13]. The region near the fusion line presents a mixed structure containing original and tempered martensite, bainite, and ferrite. The original hardness of the base metal, usually higher than 500 HV, suffers an important decrease by up to 300 HV in the heat-affected zone (HAZ). The severity of this HAZ softening effect depends on the heat input by the welding process and the heat dissipation of the material. Additionally, when a filler metal is employed, a protruding reinforcement is formed at the bead face and root. The level of stress concentration at the weld toe is responsible for an important decrease in the fatigue performance of the joint [9,14,15]. To avoid such a deleterious effect in a structural component with the aim of achieving a very high mechanical performance, the reinforcements should be carefully machined, which means involving an additional production step. Besides, there are no filler metals available in the market matching the strength level of press hardening steels. Undermatching, on the other hand, is favorable to impact the toughness of ultra-high-strength steel joints.

Laser welding mitigates or even eliminates some of the drawbacks of established electric arc welding processes as mentioned before. The extremely concentrated heat input by the laser beam permits applying an autogenous welding procedure with a very narrow HAZ. In terms of design, laser welding offers a somewhat higher level of flexibility for positioning the joints inside the structure. Furthermore, the face and root reinforcements generated by a laser welding process are comparably small so that for well-tuned laser welding conditions the subsequent flush-grinding step may not be necessary. Nevertheless, laser weld joints can contain porosity in the fusion zone (FZ) as well as the tendency of developing some degree of undercut near to the weld toe [16]. Such defects are usually tolerable in the case of conventional structural steels but may be a concern for the fatigue performance of ultra-high-strength steels.

A previous study has characterized the weld zone microstructure and static mechanical performance of dual-pass laser-welded press hardened steels of 4.5 mm gage [16]. The standard press hardening steel 22MnB5 was compared to a modified variant of similar base composition but additionally alloying niobium and molybdenum. It was demonstrated that the NbMo modified steel had a clearly finer grain structure in the HAZ and FZ of the laser welds. This reflected positively in static cross-weld tensile testing by higher tensile strength and better strain hardening capacity for the modified variant. While the standard steel exhibited a tendency towards cleavage-type failure after fracturing, the modified variant behaved more ductile. Furthermore, it appeared to be less sensitive to the presence of weld porosity. The current study investigates the performance of the same laser-welded press hardening steels under fatigue loading conditions. The aim is to provide general design data represented by S–N (Wöhler) curves for the innovative combination of heavy gauge press hardening steel and autogenous laser welding as well as to investigate the influence of the joint geometry on the fatigue performance. Furthermore, the results are expected to

reveal the potentially beneficial influence of the refined HAZ and FZ microstructure in the NbMo modified steel as all specimens have very similar weld geometry.

## 2. Materials and Methods

The chemical composition of both investigated steels is specified in Table 1. Both steels were produced on an industrial scale as hot rolled coils in quantities of several hundred tons. The steels were press hardened using water-cooled flat dies to simulate a microstructure and mechanical properties comparable to those obtained in an industrial hot stamping process. Subsequently, double-pass laser welding was performed with the as-quenched steel. Details of the welding procedure, the microstructure, and the mechanical properties are described in a previous publication [16]. The obtained mechanical properties are summarized in Table 2.

**Table 1.** Chemical composition of the investigated hot rolled 1500 MPa press hardening steels (numbers in weight percent, n.a.: not added).

| Alloy Content | C | Si | Mn | S | Cr | Mo | Ti | Nb | B |
|---|---|---|---|---|---|---|---|---|---|
| 22MnB5 | 0.22 | 0.14 | 1.19 | 0.001 | 0.22 | n.a. | 0.035 | n.a. | 0.001 |
| NbMo mod. | 0.20 | 0.19 | 1.19 | 0.002 | 0.19 | 0.157 | 0.035 | 0.04 | 0.001 |

**Table 2.** Tensile mechanical properties of as-quenched base steels and laser-welded samples transverse to weld seam (YS: yield strength, TS: tensile strength, El: total elongation).

| Properties | Base Steel (as Quenched) | | | Transverse to Laser Weld | | |
|---|---|---|---|---|---|---|
| | YS (MPa) | TS (MPa) | El (%) | YS (MPa) | TS (MPa) | El (%) |
| 22MnB5 | 1023 ± 2 | 1411 ± 7 | 9 ± 0.5 | 942 ± 91 | 1472 ± 119 | 4 ± 1.6 |
| NbMo mod. | 1075 ± 2 | 1465 ± 9 | 10 ± 0.2 | 1090 ± 111 | 1598 ± 60 | 5 ± 0.3 |

Fatigue testing was performed using a set of 25 specimens of the standard 22MnB5 steel and a set of 30 specimens of the NbMo modified steel variant. Groups of five specimens were cut out of originally laser-welded plates, so that 5 and 6 plates were used from standard 22MnB5 steel and the NbMo modified steel variant, respectively. Thus, the different groups incorporate the variability that can occur during individual laser welding runs. The oscillating fatigue load was applied transversely to the laser-welded joints. The samples were fabricated by wire electrical discharge machining to a geometry following the recommendations of the ISO 6892-1 standard (Figure 1) with a sheet thickness of approximately 4.4 mm. A careful sanding process was applied on the lateral faces of the neck and the transition radius, always in the longitudinal direction. Several finishing steps were repeated ending with 600 grit sandpaper. This finishing procedure also caused edge blunting, therefore leading to a decrease in the stress concentration factor and a lower probability of fatigue crack nucleation in the corners of the sectional area.

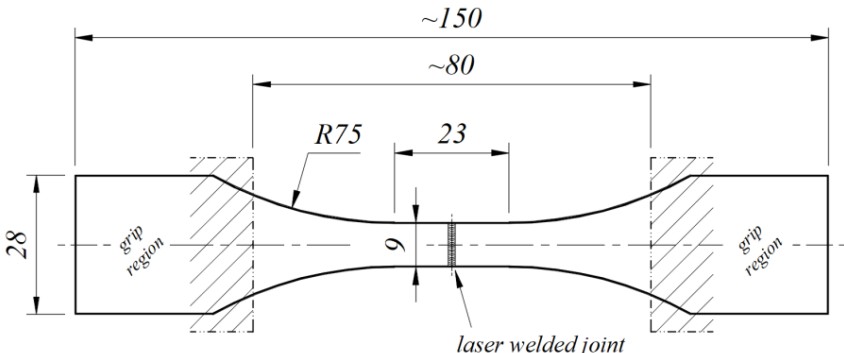

**Figure 1.** Geometry of the fatigue test specimens (all dimensions in mm) with fixture spam and grip regions according to ISO 6892-1.

Two specimens from each set of steel alloys were manually ground processed for complete removal of the weld reinforcements. To achieve a similar surface quality as would be obtained in a regular production environment, the grinding process was applied without any exceptional care, except for positioning the grinding tool in a way to align the machining grooves as far as possible along the test load direction (i.e., transversal to the welded joint). A final sanding procedure with 240 grit sandpaper was applied to the flat sample faces in a longitudinal direction.

The S–N diagrams were composed in accordance with the ASTM E739-10 (2015) standard. A load ratio of 10% was applied ($R = F_{min}/F_{max} = 0.1$). The International Institute of Welding (IIW) recommendations [14] suggest that higher test load ratios should be applied to account for the effect of residual stresses, which can induce high tensile peak values in real components and structures. However, in small-size fatigue specimens, the natural compliance of the welded plates prevents the generation of a pronounced residual stress field and, additionally, the cutting process for sample manufacturing allows stress relief. Consequently, the fatigue test results tend to present an excessive positive bias. Nevertheless, real laser-welded components usually contain compressive residual stresses in the transversal direction [17]. In view of this, lower load ratios are applied when testing laser-welded specimens [5,7,8,18].

Initially, the intention was to test at 4 specific stress ranges with values to be determined after preliminary runs. However, the scatter experienced in the related results precluded such an approach. Additionally, both the press hardening and EDM cutting processes caused small variations in the thickness and the width of the specimens, respectively. Consequently, specimens subjected to the same test load ratio presented certain variations in the stress ratio. It is important to note that the variation in stress level does not cause a relevant change in the final S–N curve. The ASTM E739 approach is very robust for dealing with this issue. Finally, changes in the stress level are commonly observed in fatigue test reports [15].

Standard engineering codes focused on the design of structures display the stress range ($\Delta\sigma = \sigma_{max} - \sigma_{min}$) instead of the nominal stress value S–N diagrams in case of welded joints. Such a procedure is adopted here because post-weld heat treatment of welded structures frequently is not viable or even possible due to technical and cost aspects. Furthermore, in the case of press-hardened steels, a stress relief treatment would generate an undesirable strength decrease due to tempering effects. Even though the effect of residual stresses in the case of laser-welded components is not as important as in conventional arc-welded joints, the weld affected region remains the prevalent failure site. Because of that, laser joints should not be expected to present a similar fatigue behavior as stress relieved weldments or the base material.

As usual, two parameters are used to directly compare results of different study cases: the stress ranges associated with two million load cycles ($\Delta\sigma_{2E6}$) and the slope of the S–N line ($m$). In agreement with reliability principles, an inferior line, associated with a conservative survival rate, is adopted in design procedures instead of the mean curve. In the present work, a survival expectance of 95% was adopted, the same level as employed in the IIW fatigue curves [14].

## 3. Results and Discussion of Fatigue Tests

### 3.1. First Run Results

The S–N diagram of the standard 22MnB5 steel is shown in Figure 2. The valid test points, used for computing the curves, are represented by open symbols. One specimen attained a very high number of cycles and was accordingly assigned as a run-out. Two flush-ground specimens were tested in a stress range $\Delta\sigma \cong 500$ MPa of which the related details will be addressed later. The obtained values of slope and stress range associated to $2(10^6)$ cycles in the 95% survival line are $m = 3$ and $\Delta\sigma_{2E6} = 91$ MPa, respectively.

As mentioned before, the fatigue damage process is multifactorial and, especially in the case of butt-welded joints, the scatter of results is noticeable. A difference in performance between welded plates (each plate generated five specimens) was observed along with the

tests. Such a difference is outlined in the diagram by assigning distinct symbols to each original plate. Table 3 summarizes the fatigue results of all individual specimens, classified according to the original plate.

**Table 3.** List of standard 22MnB5 specimens with test condition and respective results.

| Plate | $\Delta\sigma$ [MPa] | Attained Life (Cycles) | Fracture Site | Remarks |
|---|---|---|---|---|
| 1 | 300 | 205,992 | weld toe | |
| | 302.5 | 135,699 | weld toe | |
| | 360 | 181,921 | weld toe | |
| | 480.5 | 148,762 | weld toe | |
| | 482 | 30,667 | weld toe | |
| 2 | 218 | 357,902 | weld toe | |
| | 218.5 | 317,138 | weld toe | |
| | 265 | 440,407 | weld toe | |
| | 309.5 | 75,696 | weld toe | |
| | 375 | 77,238 | weld toe | |
| 3 | 204.5 | $2.25(10)^6$ ♥ | $\rightarrow$ | run out |
| | 218 | 506,041 ♥ | weld toe | |
| | 250 | 288,111 | weld toe | |
| | 375.5 | 77,026 | weld toe | |
| | 489 | 21,554 | weld toe | |
| 4 | 171.5 | 1,072,767 ♥ | weld toe | |
| | 187 | 432,672 | weld toe | |
| | 202 | 437,364 | weld toe | |
| | 263.5 | 126,215 | weld toe | |
| | 543 | 9,495 | weld toe | |
| 5 * | 298 | 205,529 | weld toe | |
| | 375 | 111,687 | weld toe | |
| | 500 | 581,294 | base metal | ground flush |
| | 502 | 62,045 | weld toe | ground flush |
| | 502 | 34,255 | weld toe | |

* 3D laser scanned specimens. ♥ Original joint configuration specimens which attained life $> 5(10^5)$ cycles to fracture.

Plates 3 and 4 exemplify the variation in fatigue performance. Specimens originating from Plate 3 contributed to a smoother S–N mean line ($m_{Plate3} > m$), while the group of specimens taken from Plate 4 dragged the mean line towards a lower level. In addition, the only run-out point corresponds to a Plate 3 specimen. On the other hand, three specimens of Plate 4 underwent a final fracture at test stress ranges than the one associated with the run out of the Plate 3 sample. Besides influencing the mean line parameters, the difference of performance also affects the variance value and, consequently, the position of the 5% and 95% survival lines. As mentioned before, design codes adopt the lower survival line as the reference for fatigue resistance. Consequently, specimens with largely deviating fatigue performance eventually have an important influence on the final resistance limit.

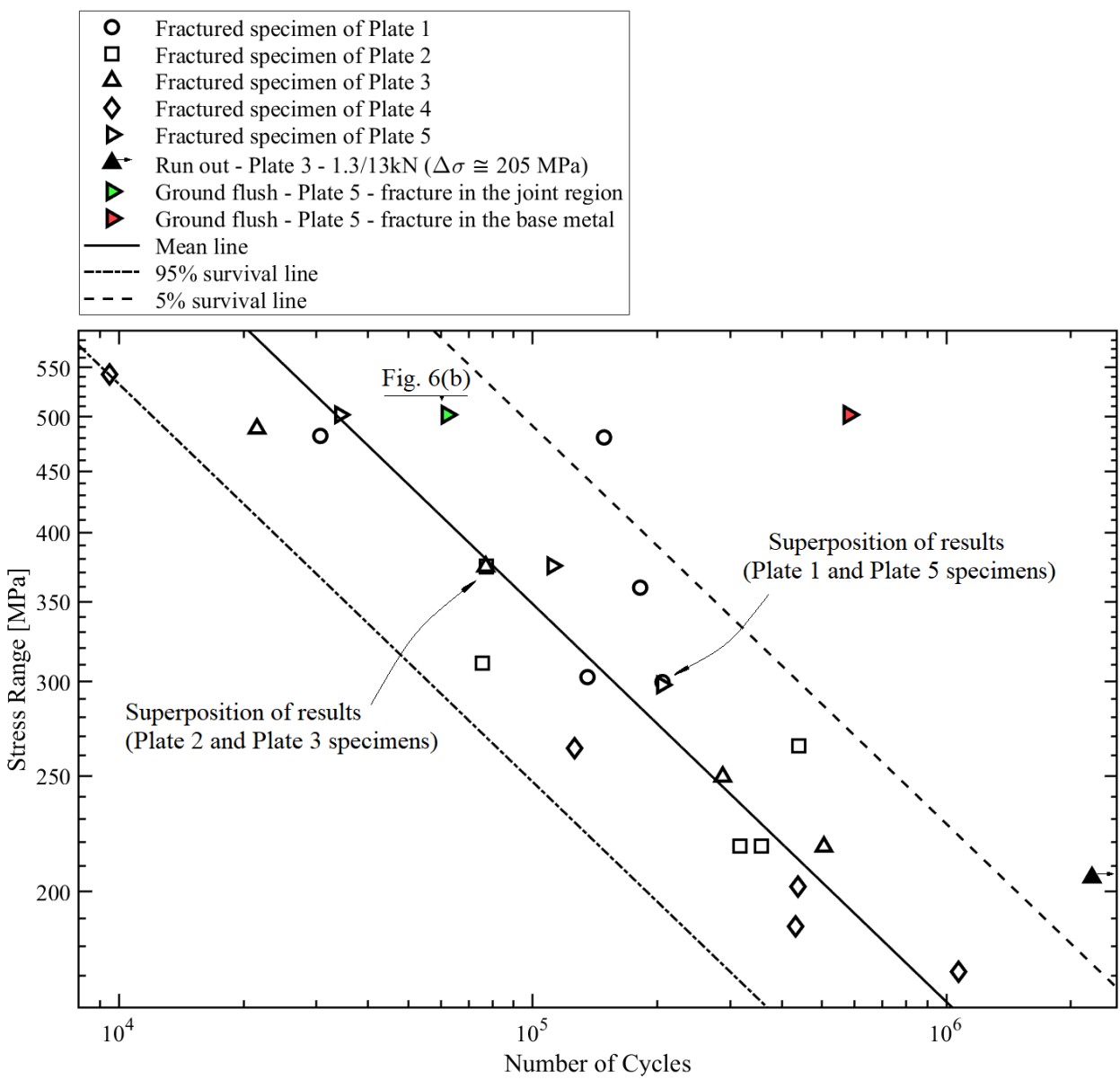

**Figure 2.** S–N diagram of laser-welded standard 22MnB5 steel.

It is also observed that most of the specimens attained a fatigue life lower than $5 \times 10^5$ cycles. Only two of the Plate 3 specimens reached either $5.1 \times 10^5$ cycles at $\Delta\sigma \cong 218$ MPa and a run out (previously defined for $>2.25 \times 10^6$ cycles) at $\Delta\sigma \cong 204.5$ MPa. One Plate 4 specimen achieved $1.07 \times 10^6$ cycles at $\Delta\sigma \cong 171.5$ MPa. Although not clearly defined at this stage, a knee point appears to exist at 500.000 cycles in the 95% survival line. The same tendency was observed in the modified alloy case, which will be presented hereafter.

The S–N diagram of the modified 22MnB5 laser-welded specimens is shown in Figure 3. Again, open symbols represent the fractured specimens and from these data, the S–N curve was computed. Specific symbols were assigned to each original welded plate. The difference in the fatigue performance of samples originating from distinct plates is also observed here. Table 4 presents a summary of the measured results covering a somewhat broader experimental base since this set contained five additional specimens. Six of the specimens attained lifetimes that were clearly longer than those expected by a 5% survival line which was a priori defined at the stage of testing, so they were assigned as run-outs. The resulting values of slope and stress range associated with $2 \times 10^6$ cycles in the 95% survival line are m = 3.86 and $\Delta\sigma_{2E6} \cong 135$ MPa, respectively. Two flush-ground specimens were tested in

a stress range $\Delta\sigma \cong 510$ MPa, which is slightly higher than the one used for the standard 22MnB5 steel.

**Table 4.** List of NbMo modified 22MnB5 specimens with test condition and respective results.

| Plate | Δσ [MPa] | Attained Life (Cycles) | Fracture Site | Remarks |
|---|---|---|---|---|
| 1 * | 307 | 216,578 | weld toe | |
| | 308.5 | 232,175 | weld toe | |
| | 368.5 | 99,760 | weld toe | |
| | 430 | 38,710 | weld toe | |
| | 491 | 28,358 | weld toe | |
| 2 | 271 | $1.5(10)^6$ ♥♣ | → | run out |
| | Retest: Δσ = 590 | 28,743♣ | weld toe | |
| | 287 | $1.5(10)^6$ ♥♣ | → | run out |
| | Retest: Δσ = 318.5 | 453,771 ♣ | weld toe | |
| | 302.5 | 253,349 | weld toe | |
| | 318 | 176,268 | weld toe | |
| | 319 | 357,346 | weld toe | |
| 3 | 287 | $1.5(10)^6$ ♥♣ | → | run out |
| | Retest: Δσ = 414.5 | 113,350 ♣ | weld toe | |
| | 303 | $1.5(10)^6$ ♥♣ | → | run out |
| | Retest: Δσ = 351 | 380,878 ♣ | base metal | |
| | 333.5 | 153,539 | weld toe | |
| | 351 | 147,028 | weld toe | |
| | 557.5 | 21,812 | weld toe | |
| 4 | 238.5 | 244,450 | weld toe | |
| | 269 | 414,152 | weld toe | |
| | 302 | 118,182 | weld toe | |
| | 303 | 141,845 | weld toe | |
| | 557 | 14,583 | weld toe | |
| 5 | 210.5 | $2.25(10)^6$ ♥♣ | → | run out |
| | 223 | $2.25(10)^6$ ♥♣ | → | run out |
| | 269 | 350,638 | weld toe | |
| | 350.5 | 99,085 | weld toe | |
| | 509.5 | 24,820 | weld toe | |
| 6 * | 302 | 800,255 ♥ | weld toe | |
| | 384.5 | 147,580 | weld toe | |
| | 510.5 | 284,941 ♣ | weld toe | ground flush |
| | 512.5 | 149,968 ♣ | weld toe | ground flush |
| | 512.5 | 47,200 | weld toe | |

* 3D laser scanned specimens. ♥ Original joint configuration specimens which attained life > $5(10^5)$ cycles to fracture. ♣ Not used for computing the fatigue resistance curves.

The apparent inflection in the curve near 500.000 cycles was also observed for this sample set, yet more pronouncedly. Out of the 22 samples fractured during the initial test run, only one reached a fatigue life of approximately 800.000 cycles with a stress range of 302 MPa. Two other samples from the same plate (Plate 6) also performed well above the expected value, while the remaining two were ground flush and cannot be directly compared. In contrast, specimens originating from the other plates, and especially from Plate 4, had obviously shorter fatigue life even when tested at similar or lower stress ranges. Such variance in performance makes the experimental planning rather difficult. Furthermore, it demonstrates that in the medium-to-high cycle life region a relatively small difference in the stress range value can cause a remarkable change in fatigue performance.

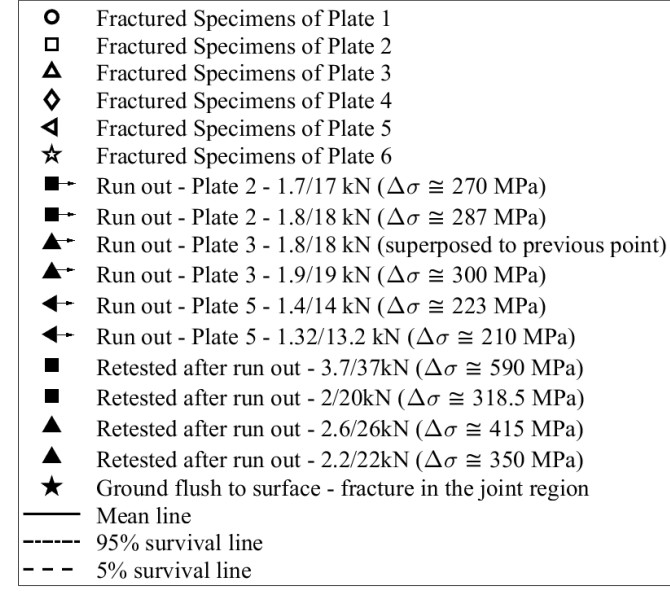

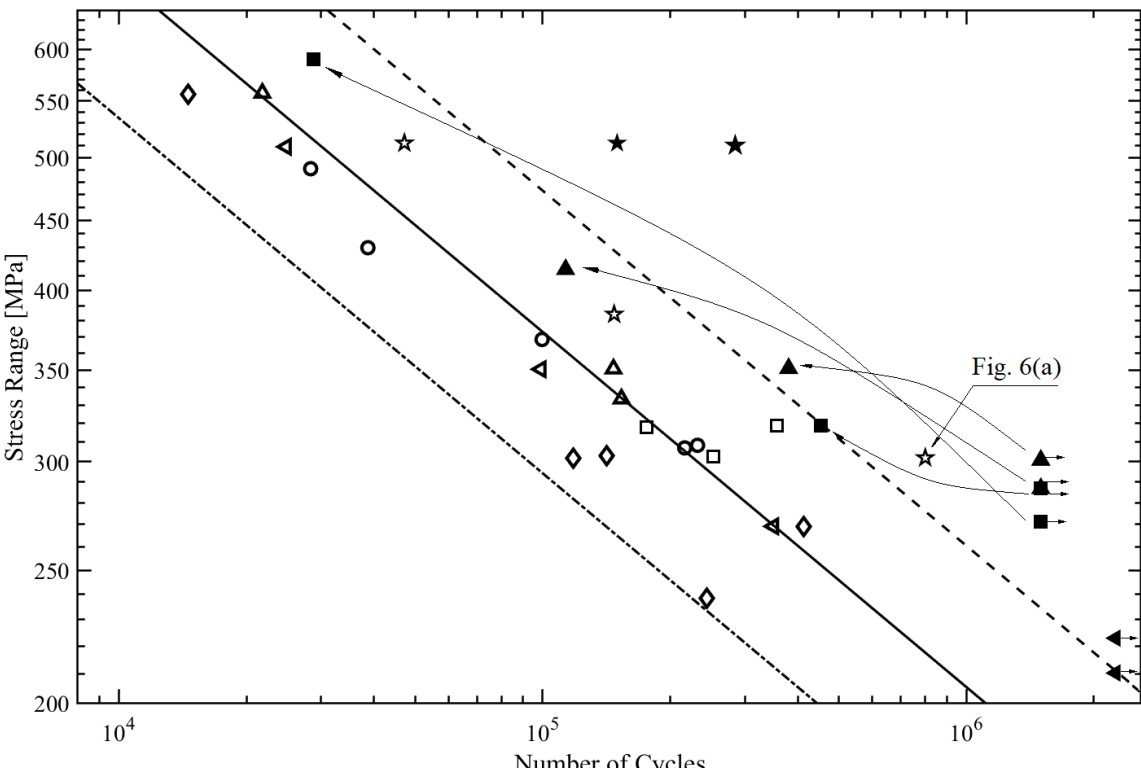

**Figure 3.** S–N diagram of laser-welded NbMo modified 22MnB5 steel.

### 3.2. Retesting of Run Out Samples

To expand the analysis, four run-out specimens originating from Plates 2 and 3 of the NbMo modified steel were retested at higher stress ranges than in the first run (indicated by arrows in Figure 3). However, these test results were not considered when computing the S–N curves. The retested specimens showed superior fatigue performance in comparison to the entire first run test population. Even though the number of retested samples is too small to infer a statistically sound conclusion, two hypotheses were established that could be responsible for this superior fatigue performance.

The first one relates to the natural scatter associated with the laser welding process. Taking into consideration that all the specimens of Plates 2 and 3 exhibited a medium to

high fatigue performance, it is reasonable to assume that the run-out specimens were of intrinsically superior quality. In other words, if these specimens had been originally tested at the larger stress ranges adopted during retesting, they would have generated similarly good results.

The second hypothesis relates to the fact that the tested materials present a very high mechanical strength. Consequently, the low stress ranges applied to these samples in the first run, could have induced some level of stress relief instead of promoting crack propagation, leading to better fatigue performance during retesting. Previous work by de Almeida et al. [16] indicated the presence of a reasonable level of ductility in the HAZ, especially in the NbMo modified steel.

According to Zerbst et al. [19], plastic deformation is the basic mechanism of residual stress relief in any case. The low cycle fatigue approach usually addresses specific cases, where great strain values are generated by extreme loads. In these cases, the strain spreads over the whole critical section of the structural part. However, in the case of welded joints, small and localized plastic deformation occurs even in the case of high cycle lives, attained as a result of relatively low-stress ranges. Welded joints naturally represent deviations on the load path, which means that a stress concentration factor is associated with the weld toes. Additionally, peaks of tensile residual stress occur in specific regions of the weld joint. Thus, the superposition of effects may cause this region to suffer very localized plastic deformation consequently facilitating residual stress relief. The convergence of results of defective and non-defective specimens in the high cycle fatigue region reported by Ottersböck et al. [20], as well as the residual stress mitigation, verified with the X-ray diffraction technique by Pessard et al. [4] additionally support this hypothesis. Notwithstanding the importance of these aspects, the verification of the above-described hypotheses was not further pursued in the context of this study.

### 3.3. Comparison of Standard 22mnb5 and NbMo Modified Steels

Figure 4 shows the superposition of the 95% survival lines derived from the fatigue tests of both steels. All data points originating from first-run tests are also included in this representation. The 95% survival line related to the NbMo modified steel has a smaller slope and thus predicts increasingly higher stress ranges with the number of cycles. The data points of the standard 22MnB5 steel reveal an obviously larger scattering range. The six run-out data points of the NbMo modified steel scatter between a stress range of 210 and 303 MPa whereas the single run-out data point of the standard steel is situated at a lower stress range of 204.5 MPa.

The IIW FAT90 fatigue curve, designating high-quality butt-welded joints with a reinforcement height lower than 10% of the sheet gauge and linear plate misalignment less than 5% of the sheet gage is also plotted in the diagram. That fatigue curve is associated with a 95% survival rate representing slope values of m = 3 and m = 22 for the high and very high cycles to failure regions, respectively. These two regions are separated by a knee point at N = $10^7$. The 95% survival line generated by the standard 22MnB5 samples exactly matches the IIW fatigue curve in the high cycle region as it has the same value of m = 3 while that of the NbMo modified steel is on a superior level with m = 3.86.

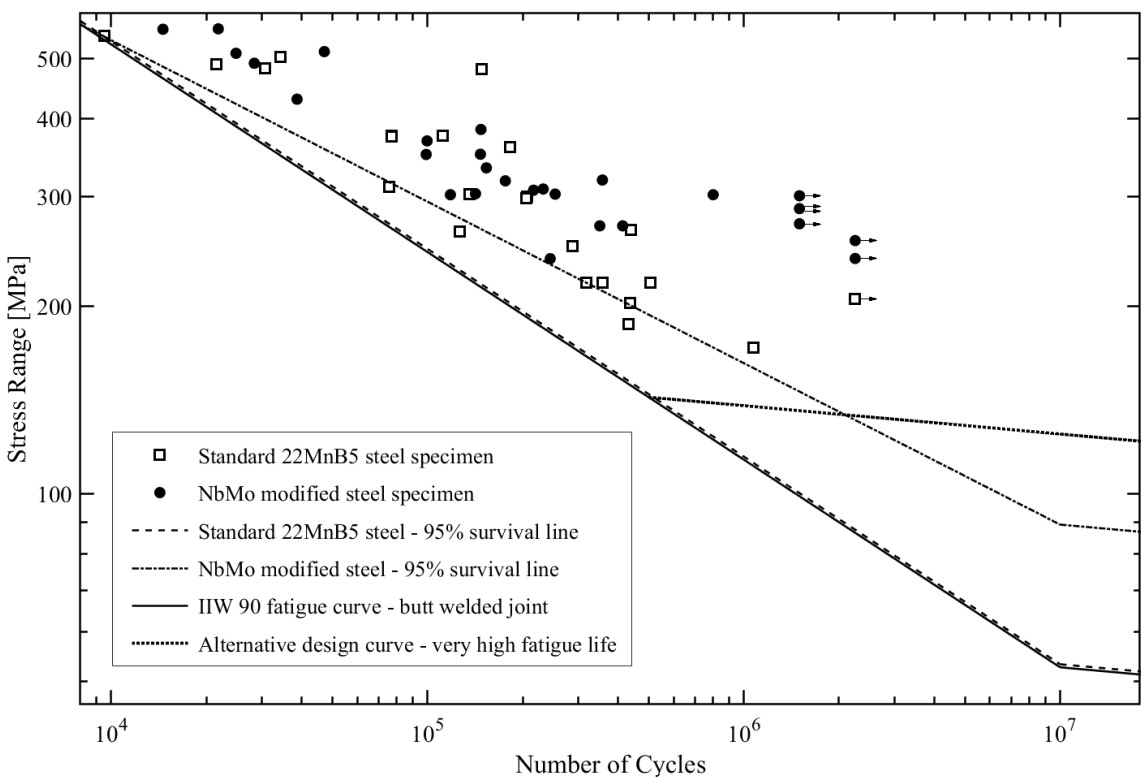

**Figure 4.** Comparison of S–N diagrams for the two laser-welded 22MnB5 steels.

### 3.4. Analysis of Fatigue Fracture Surface

Figure 5 compares the fracture surfaces of the standard and NbMo modified steels of specimens tested at similar stress ranges of around 309 MPa. The ratio of maximum test stress of around 344 MPa to the ultimate tensile strength of the base metals being 1411 MPa for the 22MnB5 steel and 1465 MPa for the NbMo modified steel is $0.24 \cdot S_{u\text{-standard}}$ and $0.23 \cdot S_{u\text{-modified}}$, respectively. The fatigue cycles reached under these conditions are 75,696 for the standard steel and 232,175 for the NbMo modified steel.

The large crack-tip shown in Figure 5 indicates that several intrusions propagated and encountered to form a unique crack front. It is important to mention that the test stress ranges adopted were relatively high because the base metal and most of the heat-affected zone presented remarkable mechanical strength. Thus, sufficient energy was provided for several intrusions to propagate. It is not supposed that a laser welding process would generate a greater number of intrusions than a conventional process. Accordingly, such small crack-like flaws are considered responsible for triggering the damage process, completely suppressing the crack nucleation period. The matching of the standard 22MnB5 fatigue line with the IIW 90 curve, representing the fatigue resistance of high-quality arc-welded joints in the case of conventional and high strength structural steels, indicates that the ultra-high-strength attained by the press hardening process does not translate into a correspondingly better fatigue performance. On the contrary, the NbMo modified steel allowed exploiting the higher base metal strength to achieve superior fatigue performance. Since the laser welding parameters were identical for both steels, the performance improvement must be related to the microstructural differences in the HAZ and FZ, as will be discussed in Section 4 of this paper.

Maddox [9] observed that very narrow butt-welded joints, such as those formed by laser beam welding or similar techniques, usually provide a remarkable fatigue performance, analogous to the case of high-quality arc-welded joints. Such a performance would stem from the minor size effect associated with the reinforcements, which are so small that they cause no relevant load path deviation. In other words, the welded joint stress-concentration factor is close to unity.

Nevertheless, a fracture process triggered by the coalescence of several small cracks and the salient difference of performance depending on the original welded plates observed in the present tests indicated that the stress concentration at the weld toe was an essential aspect to be analyzed. Butt-welded joints may present a remarkable scatter in fatigue test results, depending on reinforcements geometry and misalignment aspects [7–9,18,20]. Therefore, some specific investigations were incorporated in the present work schedule for clarifying the influence of these aspects.

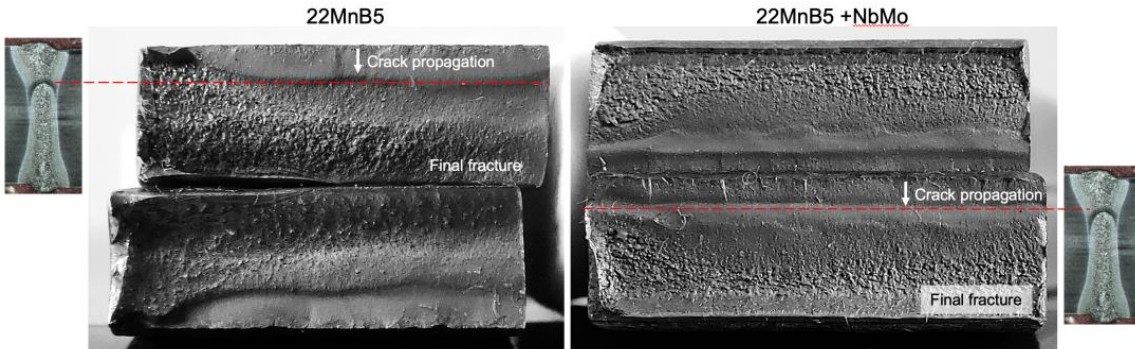

**Figure 5.** Mirrored fracture surfaces after fatigue failure. Standard 22MnB5 steel, tested at $\Delta\sigma$ = 309.5 MPa [$\sigma_{max} \cong$ 344 MPa $\cong$ (0.24)$S_{u\text{-standard}}$] failure after 75,696 cycles. NbMo modified 22MnB5 steel, tested at $\Delta\sigma$ = 307 MPa [$\sigma_{max} \cong$ 341 MPa $\cong$ (0.23)$S_{u\text{-modified}}$] failure after 216,578 cycles. Representative cross-sectional micrograph of the double-pass laser weld indicates the position of the weld overlap region (red dashed line).

*3.5. Analysis of Joint Geometry Effects*

Two specimens of each steel group were flush ground before fatigue testing to completely remove the weld reinforcements and toes. The obtained data of these samples are represented by full symbols in the respective S–N diagrams of Figures 2 and 3. A superior fatigue performance with respect to the mean line is evident. Three of the specimens surpassed the respective 5% survival lines, with one sample unexpectedly fracturing away from the heat-affected zone at the frontier between the neck and the transition radius. Clearly, the geometry improvement provided by the grinding process generated a relevant decrease in the stress concentration level at the weld toes, causing the test conditions to be different from those applied in the original set of specimens. Consequently, the ground flush results were not used for computing the reported S–N curves.

Five samples of the standard steel (Plate 5) and 10 samples of the NbMo modified steel (Plates 1 and 6) were measured by laser-based 3D scanning (Digimill 3D model by Tecnodrill, Novo Hamburgo, Brazil) based on conoscopic holography allowing a depth resolution of up to 0.01 mm. Surface profiles were extracted from the measured 3D data sets in a transverse direction to the laser weld at several positions (Figure 6). The parameters evaluated from such profiles were the reinforcement height, *h*, the eccentricity value, *e*, flank angle, *θ*, and the misalignment angle, *α*. Generally, the fatigue samples taken out of the two 3D scanned plates showed good fatigue performance (Figures 2 and 3) indicating that both plates presented median or high joint quality levels. For further discriminating the relevance of geometrical influences, two samples with pronounced differences will be discussed in the following sections.

Figure 6a shows the laser weld profile of a sample extracted from plate no. 6 (NbMo modified steel) reflecting a reinforcement height-to-plate thickness ratio, $h_{total}/t = (h + e)/t$, of 0.078. This value is below the limit of 0.1 assigned to high-quality joints according to the IIW recommendations [14]. Lillemäe et al. [18] outlined the effectiveness of the flank angle *θ* as a joint geometry parameter. The ISO 5817 code adopts the following classification limits as $\theta \leq 90°$, $\theta \leq 70°$ and $\theta \leq 30°$ to low, regular and high-quality joints, respectively [18]. Accordingly, the measured flank angle value of 53.5° is classified as the weld geometry being of regular quality. No remarkable angular misalignment is observed in the profile

while the eccentricity is very subtle. The misalignment ratio *e/t* of 1.4% is clearly below the 5% limit corresponding to high-quality joints.

Figure 6b shows a laser weld originating from Plate 5 (standard 22MnB5). It has a small reinforcement height and lower flank angle than the previous sample. However, the sample was flush ground before fatigue testing so these features are irrelevant to the fatigue performance. Concerning other features, the cross-sectional profile exhibits a small undercut as well as a more pronounced eccentricity and misalignment angle.

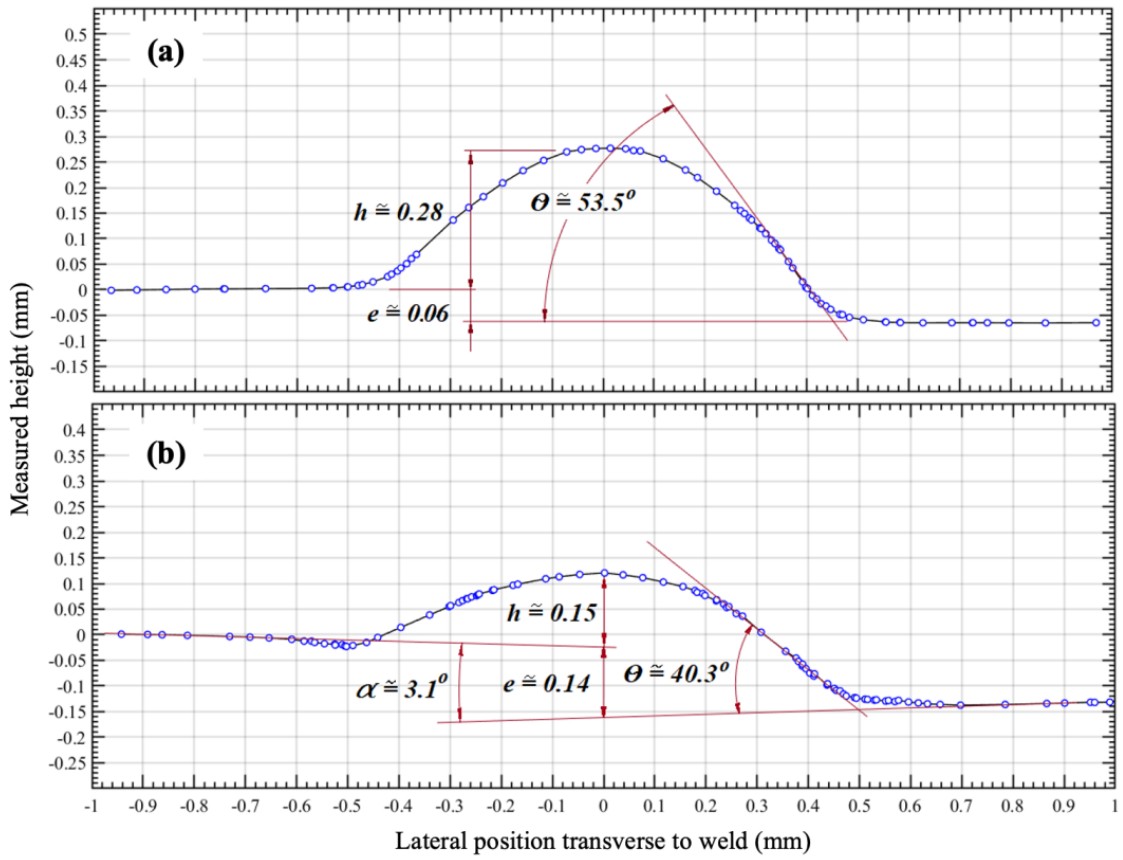

**Figure 6.** Profilometric measurements across selected laser weld seams: (**a**) NbMo modified steel specimen and (**b**) standard steel specimen before flush grinding.

Flush grinding as such would promote the sample into the high-quality region of the S–N diagram. Nevertheless, this sample performed with $6.2 \times 10^4$ cycles at a stress amplitude of $\Delta\sigma \cong 502$ Mpa, clearly worse than all the other flush ground samples tested. Work by Lillemäe et al. [18] suggested that small and shallow undercut defects such as the one seen in Figure 6b do not provoke a significant decrease in fatigue performance. Thus, this can not explain the underperforming fatigue behavior of this sample.

The more pronounced eccentricity and misalignment observed in this welded sample generate secondary moments under the oscillating fatigue load which, in turn, alter the stress level in the weld toe regions. Accurately determining the real stress amplification caused by angular misalignment is a complex task. As described by Lillemäe et al. [8], the misalignment angle attained in the welding process changes when the test machine grips are actuated. In other words, fixing the specimen into the grips tends to straighten the sample but generates additional stress components acting on the weld toes. Concerning angular misalignment, complete congruency between laboratory fatigue test results and the fatigue performance of a welded component under real circumstances is not expected, mainly because of boundary conditions and body compliance differences.

Concerning misalignment issues, Lillemäe et al. [8] employed the structural hot spot and the notch stress assessment methods for computing stress magnification and notch

factors in case of complex welded joints with relevant geometry defects. Alternatively, the thorough study on butt-welded joints reported by Fricke et al. [7] was based on the very reasonable premise that misalignment errors can be hardly controlled under field and shop floor conditions so that a notable scatter of results is unavoidable. In that context, the fatigue performance was directly correlated with high, regular, and low joint quality levels. This second approach was adopted to correlate the results of the present work.

Referring to Figure 6a,b, the welded joint of the NbMo modified steel sample achieved a regular quality level in terms of reinforcement shape, however, a high-quality level regarding misalignment. This combination of geometry features resulted in an excellent fatigue performance. On the contrary, the flush ground sample of the standard steel was on a superior level concerning reinforcement geometry but comprised evident misalignment deviation. In this case, a poor fatigue performance was attained when compared with the rest of the specimens in the test group. These observations to some degree corroborate the statement that in laser-welded joints the reinforcement dimensions are too small to cause relevant size or notch effects Maddox (2002). However, the results also indicate that an excellent fatigue performance of welded press-hardened steel joints is only attained if the misalignment deficiencies remain within tight quality limits.

The lower scattering in the fatigue results of the NbMo modified steel implies that this material was apparently less susceptible to producing misalignment considering that the laser welding conditions and edge preparation were identical to those of the standard steel. The specimens of the standard 22MnB5 steel presented greater eccentricity and angular misalignment values than compared to the NbMo modified steel. Other joint geometry aspects, such as the reinforcement height, were almost the same in both groups of specimens. Thus, the previously reported [16] positive effect of the addition of Nb and Mo on the HAZ microstructure (see also Section 4 below) apparently also improves the geometric quality of the welded samples as reflected by Figure 6.

Ultra-high-strength steels, in the absence of significant geometrical and material defects, may provide remarkable fatigue performance. However, the fracture surfaces exposed in Figure 5 undoubtedly prove that intrusions generated during the welding procedure were responsible for triggering a severe damage process which, in turn, strongly limited the number of cycles to fracture of the specimens. Flush grinding in most cases promoted a relevant increase in fatigue life indicating that the notch effect associated with the reinforcement is not negligible, even in the exceptionally narrow joints generated by the laser welding process. However, a comparison of the weld bead profiles suggests that linear and angular misalignments apparently have a more prevalent contribution to poor fatigue performance and high scatter.

*3.6. Transition to Fatigue Limit*

The data points in Figure 4 indicate that geometry-related effects mainly dominate the fatigue performance in the region below $5 \times 10^5$ fatigue cycles. Therefore, an additional line with the slope value m = 22 (representing the extreme fatigue life region according to IIW recommendations) and a knee point at $5 \times 10^5$ cycles was superposed to the FAT 90 resistance curve shown in Figure 4. The stress range associated with $5 \times 10^5$ cycles is $\Delta\sigma_{5E5} \cong 143$ MPa, corresponding to a peak stress value $\sigma_{5E5max} \cong 159$ MPa or approximately $0.11 \cdot S_u$ of the base material. This constructed S–N curve complies well with the results obtained from both steels.

The adoption of a positive load ratio in the applied fatigue tests strongly limited the possible occurrence of crack closure. However, the combination of the ultra-high-strength capacity of press-hardened steel with a potential residual stress relief mechanism occurring at lower test stress ranges may be responsible for the observed earlier onset of the extreme fatigue life regime at $5 \times 10^5$ cycles. For conventional and high strength structural steels such a transition usually occurs between $10^6$ and $10^7$ cycles and a notable increase in fatigue performance, from a specific number of cycles onwards, is unusual. The fatigue performance increase in the present case may result from: (i) the remarkable mechanical

strength of the base metal, (ii) the superior quality provided by the laser welding process, or (iii) the sum of both aspects. Evidently, the available results serve only as a primary indication. Future investigations could address the observed phenomenon of an increment of fatigue life from $5 \times 10^5$ cycles onwards clarifying some of the lacking aspects.

## 4. Microstructural Influences

The observed performance differences of the two steels in the fatigue behavior and the analysis of influences related to the weld geometrical imply that the microstructural differences in the HAZ must play an important role. Figure 7 gives an overview of the microstructural characteristics in the heat affected zone of laser weld seams of both steels. The hardness distribution and configuration of the HAZ are identical. The base material (BM) hardness of 550 HV is typical for die quenched steels with a base composition of 22MnB5. In the fusion zone (FZ) and the heat-affected zone (HAZ) between the fusion line (FL) and the $A_3$ limit, the hardness is approximately 50 HV higher than in the base material due to the significantly higher cooling rate acting on the fully austenitic microstructure immediately after laser welding as compared to die quenching of the BM.

The hardness trough extending from the $A_3$ limit towards the BM reaches a minimum of about 350 HV just outside the $A_1$ limit. This area corresponds to a highly tempered martensite, typically comprising excellent toughness. In the intercritical range between $A_1$ and $A_3$, a mixed microstructure of recrystallized ferrite in combination with martensite and bainite prevails [16]. Estimating the tensile strength at around 1050 MPa at the position of lowest hardness and assuming that the yield-to-tensile ratio is not less than 0.6 in the intercritical range, even the lowest yield strength in the HAZ is expected to remain above 630 MPa. In comparison, the highest peak stresses applied in the current fatigue tests are below that value (except for one sample re-tested at $\Delta\sigma$ = 590 MPa).

The tensile data (Table 2) indicate a higher tensile strength of the welded samples as compared to the base material despite the localized soft zone. However, the yield strength of 22MnB5 after welding is lower, while it is similar to the base material for the NbMo modified steel. It is not expected that the softest zone in the HAZ significantly influences the macroscopic strength since its dimensional extension is extremely narrow and it is constrained by a much stronger microstructure on either side. The soft zone, due to its complex phase characteristics, shows very efficient work-hardening while any transverse contraction is constrained by the neighboring stronger microstructures [16]. The fusion zone in the middle of the tensile sample has the highest hardness and impedes the transversal contraction behavior during the tensile deformation. However, it should be emphasized that pre-existing crack-like defects are considerably more relevant regarding fatigue performance than the variations in mechanical strength observed in different regions of the heat-affected zone.

The fully martensitic microstructure in the HAZ between FL and the $A_3$ limit was characterized by reconstructing the prior austenite grain (PAG) structure from electron backscattering diffraction (EBSD) analysis [de Almeida]. It is obvious that the NbMo modified steel has a much finer PAG size, more equiaxed in the interpass zone (HAZ IP) where the first and second weld passes overlap. In the HAZ generated by a single weld pass (HAZ P1), the PAG morphology is generally equiaxed but still finer sized in the NbMo modified steel. It was demonstrated by Thompson et al. [21] for a modified 8620 steel with rather similar base chemistry as the present steels that Nb microalloying and the resulting PAG size refinement leads to improved fatigue behavior. The endurance limit stress thereby showed a reasonably good correlation with the inverse square root of the PAG size such as a Hall–Petch type relationship. Moreover, the steels with finer PAG size endured a higher stress amplitude at a given number of fatigue cycles in the low cycle region. This reported PAG size effect is principally applicable to the present steels, only that the microstructure in the weld seams is clearly more complex and inhomogeneous. Furthermore, it is reasonable to assume that a coarse and elongated PAG morphology causes higher residual stresses upon quenching into martensite. This is related to the

dilatation upon the martensitic transformation of the austenite grains. A finer-grained and equiaxed PAG microstructure reduces the magnitude of the transformational dilatation and statistically offers a better chance to redistribute or compensate the resulting stresses. The presence of significant residual stresses is macroscopically reflected in the angular misalignment of the laser-welded samples (Figure 6b). It has been already discussed in paragraph 3.5 of this paper that misalignment can deteriorate fatigue performance. Future work incorporating detailed fractography and residual stress analysis is expected to reveal the influence of these microstructural features in more detail.

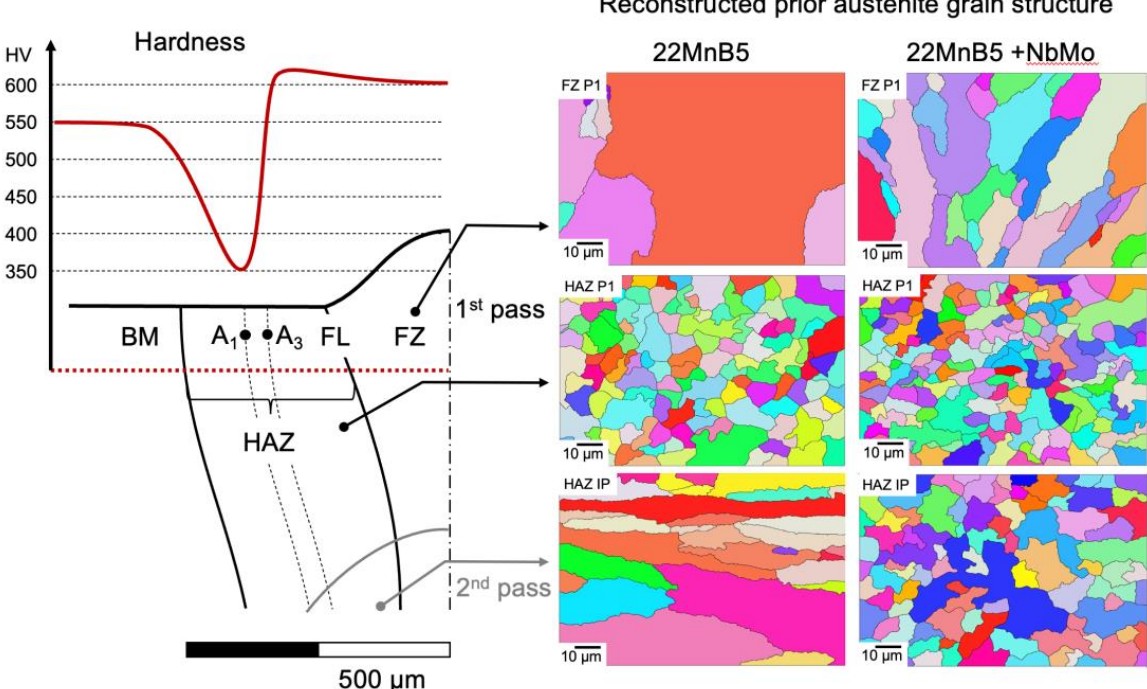

**Figure 7.** Microstructural characteristics in the heat-affected zone (HAZ) and fusion zone (FZ); $A_1$ and $A_3$ are the transformation temperatures delineating the intercritical zone (dashed lines); IP designates the interpass zone experiencing two successive weld passes; BM is the base material. Micro Vickers hardness evolution across the weld zone (dotted red line) is similar in both steels.

## 5. Conclusions

The fatigue behavior of autogenous laser butt-welded joints in two different hot rolled press hardening steels (22MnB5) of 4.5 mm gauge were presented and discussed. It appears that the IIW 90 fatigue curve, which is associated with high-quality butt joints, represents a conservative approach to the observed performance of both steels. In addition to this principally good result, the samples of the NbMo modified alloy still showed a further improved S–N mean line as well as a lower scatter band than the standard 22MnB5 steel. The careful consideration of relevant influencing factors suggested that the more favorable microstructure, i.e., finer prior austenite grain size with more equiaxed morphology, in the heat-affected zone of the NbMo modified press hardening steel should be responsible for this performance increase. With these insights, further investigations can be designed to reveal a detailed understanding of the interaction of damage mechanisms and HAZ microstructure of such laser-welded press hardening steel including a thorough analysis of the potential influence of residual stresses.

**Author Contributions:** Formal analysis, drafting the manuscript, reviewing and editing: D.T.d.A. and J.H.C.d.S.; formal analysis, supervision: T.G.R.C.; formal analysis: K.E.B., F.P.d.S., M.S.F.d.L.; formal analysis, writing—review and editing: H.M. All authors have read and agreed to the published version of the manuscript.

**Funding:** This research received no external funding.

**Institutional Review Board Statement:** Not applicable.

**Informed Consent Statement:** Not applicable.

**Data Availability Statement:** Not applicable.

**Acknowledgments:** The support of CBMM to this project is gratefully acknowledged.

**Conflicts of Interest:** The authors declare no conflict of interest.

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
