# Peer review of "Fatigue Performance of Laser Welds in Heavy-Gage Press Hardening Steels"

_metals, doi:10.3390/met12040580_

Round 1
Reviewer 1 Report
In this paper, the authors studied the fatigue performance of laser welded joints of two press hardening steels, i.e., a standard 22MnB5 steel and a steel modified with Mo and Nb. The results show the fatigue performance of laser welded joints of the MoNb modified 22MnB5 steel is better than the standard 22MnB5 steel. Meanwhile, the joint geometry of welded joint and mechanical properties of HAZ and FZ have a certain effect on the fatigue performance of welded joint. All results indicate that laser welded joints of press hardened steels could offer a very favorable fatigue performance being considerably better than that of conventional and high strength structural steels.
The topic is very interesting. Nevertheless, the performed experiment and analysis do not appear exhaustive. Therefore, in referee’s opinion, the paper can be accepted with major revision.
- The fracture surface is considered to clearly reveal the sequence of fatigue fracture. Although some fracture surfaces are shown, there are only microfracture morphology, a more detailed analysis is not given, especially the lack of microscopic analysis. It is difficult to reveal the effect of microstructure on fatigue fracture. Therefore, more fracture surfaces should be observed.
- it is well known that the geometry of joint strongly affects the fatigue performance, the stress concentration caused by geometry of joint should be calculated and analyzed in detail, and its effect on fatigue performance should be discussed.
- The microstructure of mechanical properties of HAZ may strongly affect the behavior of fatigue crack initiation and propagation, moreover, affect the fatigue performance. In Fig. 7 shows that the microstructure of HAZ of two steels is obviously different. The effect of microstructure on hardness and behavior of fatigue crack initiation and propagation must be analyzed.
Reviewer 2 Report
Congratulations to the authors for the excellent research they have done. However, laboratory work is very difficult, time consuming and costly. I believe that valuable results have been obtained and can be used in the industry after the completion of the testing process because it is acceptable to the industry as a general idea with the results of the initial review. In the following, some parts of the article need to be corrected, and in some cases, it is necessary for the authors to add more explanations. Some tips to increase the quality of the manuscript are:
1-Page 3 line 112, why use different number of samples for two materials (25 and 30)?
2- why the fatigue tests were performed under positive stress ratio (R=0.1), While to extract the S-N curve, completely reversed loading is usually used (R=0). Is there a specific standard for this issue? I know that the authors referred to Ref No. 5, 7, 8, and 18 for using lower load ratios for testing the laser welded specimens.
3-Related to figures 2-4, horizontal lines inside the image should be removed for a clearer view of the data. The marker size should be larger. Finally, replace them with the higher quality images.
4- Related to Tables 3 and 4, why do not use the same stress range for testing the samples? what is the meaning of expected life? How did the authors make this prediction? Using which tool? It is better to add images of failed specimens in the remarks column to show that the failure occur in weld toe.
5- What was your criterion for selecting the stress range value in retested samples, for example: table 4, plate 3 and stress range of 414.5 MPa?
6- Page 8 line 245, should check it with crack inspection via NDT methods such as eddy current, thermal photographic or ....
7- Page 8 line 248, the authors stated that localized plastic strain occur, it means that there is low-cycle fatigue and should use strain-life approaches, please describe more about this subject to avoid confusing the reader.
8- Page 9 line 278, the phrase "Figure 5 Figure compares ....." should change to "Figure 5 compares ..."
9- Page 10 lines 323-326, it should be noted that welding process create tensile residual stress in the surface and the toe area. so, by using grinding the toe, you delete this residual stress and the fatigue life improved. you should extend this section of the paper around this subject.
10- Page 12 lines 365-369, in the axial testing machine, you sould use the support metal plate on both sides of specimen as a proper fixture and after that fixing specimens on the grips that can remain the misalighnment angle and do not strightened it.
11- Page 12 lines 390-393, I do not draw such a conclusion from the presented material and I strongly suggest that you add more details.
12- Section 3.3 should be deleted and I believe that have no enough data and it is unnecessary.
13- Page 13 lines 429-430, it is not true. for example, to improve the tensile strength of a material, i can cut and weld it by laser? In addition, I see the data in the Table 2, in this case, the materials should be failed and not weld zone.
14- In the conclusion section, I believe that lines 501-512 should be deleted. 15- Literature review should be extended using more papers published recently.
Reviewer 3 Report
The article highlights peculiarities of the properties (strength and fatigue life) of laser welded joints of two press hardening steels: a standard 22MnB5 and a variant modified by a combination of niobium and molybdenum (NbMo) alloying. However, insufficient attention is paid to microstructural and microfractographic studies. Also, no attention is paid to the effect of residual stresses that occur in the welding process.
The article is interesting, but a number of shortcomings need to be corrected:
- Please explain why the samples obtained from different plates were separated. What is the difference between the plates? The results obtained on different plates must be substantiated.
- In Table 3, please remove the duplicate values of stress (column 3), which are already given in column 1.
- 5 is not informative. The authors point out (Lines 286-287) the following: “Therefore, such small crack-like flaws are considered responsible for triggering the damage process…”. However, this cannot be recognized in Fig.5. Fractography analysis of both steels (Lines 452-465) based on Fig.5 is practically impossible. Such an analysis should be performed at higher magnifications.
- The results presented in Table 2 and Fig. 7 should be explained. The welded joint has higher strength compared to the base steel (Table 2). However, according to Fig. 7, the heat affected zone has the lowest hardness (hence strength). How the authors determined the characteristics of the welded joint, which are higher compared to the base steel and the heat affected zone? The sample will break during the tests in the least strong place (in this case in the heat affected zone or the base steel).
- In Fig.7, the scale bar cannot be recognized.
- It would be advisable to estimate the residual stresses in the welded joint samples of both steels and take into account their effect on fatigue life.
- Please shorten the conclusions. Conclusions should be short and concise.
Round 2
Reviewer 2 Report
The authors tried to provide the revised manuscript based on the reviewers' comments. They also responded to the comments separately and it is acceptable. However, it is necessary to add the equation used for the fatigue life prediction values with a brief explanation and appropriate reference to the text (see the answer of remark 4).
Reviewer 3 Report
The authors took into account the comments of the reviewer and made appropriate corrections to the manuscript. The article is interesting and can be accepted in the present form.